# GOOD: DECODING-TIME BLACK-BOX LLM ALIGNMENT

## ABSTRACT

Large Language Models (LLMs) have demonstrated immense potential across various applications. However, aligning these models with specific real-world tasks and human preferences typically requires resource-intensive fine-tuning processes such as Supervised Fine-Tuning (SFT) and Reinforcement Learning from Human Feedback (RLHF). In this paper, we propose GOOD (Guided Online Optimal Decoding), a novel alignment method that enhances pre-trained models at decoding time without requiring access to their parameters or vocabularies. We observed that different aligned models exhibit similarities in their decisions of alignment-related tokens. Inspired by this, GOOD utilizes a pair of guiding models to identify critical positions related to alignment and adjusts the model's output dynamically during the decoding phase. Notably, the interaction between the guiding models and the guided model occurs at the string level, enabling GOOD to be applied to align even black-box models with different vocabularies. Experiments show that in weak-to-strong alignment, GOOD can achieve performance comparable to direct fine-tuning in terms of comprehensive capability and harmless generation, reaching relative scores up to 102% and 99% without sacrificing decoding efficiency. Even when guiding across model families, it can achieve 98% and 103% of the target performance on the two tasks, respectively. Moreover, GOOD can be applied to enhance already aligned models (improving pass@1 by 52% in code enhancement), making it compatible with various existing alignment techniques.

## 1 INTRODUCTION

Large Language Models (LLMs) have demonstrated remarkable potential across various applications, such as programming, writing, language translation, etc. After pre-training on a huge amount of text corpus, they often require further alignment to adapt to specific real-world tasks as well as human values and preferences. The alignment process usually involves Instruction Tuning (Wei et al., 2021) and Preference

Table 1: Comparison of GOOD and other representative tuning-free alignment methods (Note: "🟢" indicates DO NOT NEED, "🔴" indicates NEED).

| Method | Need extra training? | Need special designed prompt? | Need accessing vocabulary and logits of target model? | Need extra test-time computation? |
|---|---|---|---|---|
| URIAL | 🟢 | 🔴 | 🟢 | 🔴 |
| RAIN | 🟢 | 🟢 | 🟢 | 🔴 |
| GenARM | 🔴 | 🟢 | 🔴 | 🔴 |
| Proxy-tuning | 🔴 | 🟢 | 🔴 | 🔴 |
| Aligner | 🔴 | 🟢 | 🟢 | 🔴 |
| GOOD | 🟢 | 🟢 | 🟢 | 🟢 |

Learning (Ouyang et al., 2022), typically implemented through methods such as SFT (Wei et al., 2021) and RLHF (Ouyang et al., 2022). These alignment methods have significantly enhanced the capabilities of LLMs, suggesting that alignment-related tuning is crucial for developing AI assistants (Bubeck et al., 2023).

However, fine-tuning-based alignment methods have three major problems. (1) They are resource-intensive, requiring extensive training data and substantial computational power. (2) The same fine-tuning process is often applied to different models, leading to redundancy. (3) They necessitate direct access to the model's parameters, which is often impractical for state-of-the-art closed-source models (e.g., GPT-4o (OpenAI, 2024)).

Given these challenges, there is a growing interest in alignment methods that do not require fine-tuning. Zhou et al. (2024) proposed the *Superficial Alignment Hypothesis*, suggesting that most of a model's knowledge and capabilities are acquired during pre-training, with alignment primarily teaching the model which sub-distribution of responses to utilize in user interactions. Building

on this premise, recent work such as URIAL (Lin et al., 2023) has analyzed token shifts between pre-trained LLMs and their aligned counterparts, finding that most token distribution changes occur in language style-related tokens (e.g., discourse markers, safety disclaimers). RAIN (Li et al., 2023) attempts to use the pre-trained LLMs to evaluate their own generation and use the evaluation results to guide rewind and generation for AI safety. Liu et al. (2024) proposed Proxy-Tuning, which achieves an alignment effect similar to direct fine-tuning by computing the logits difference between the pre-trained model and its aligned version, then applying this vector to the output logits of another model in the same model series.

Nevertheless, these tuning-free alignment methods face several limitations that restrict their application across diverse scenarios. (1) Specifically designed in-context prompts cannot fully reflect user preferences across different scenarios, hence failing to adapt flexibly to various types of task. (2) Methods that adjust decoding based on token logits are constrained by the model's vocabulary, limiting their use to within the same model series. (3) Additionally, existing methods typically incur additional test-time computational costs, rendering them less economically viable. These challenges significantly hinder the practical utility of current non-tuning alignment methods, emphasizing the need for more adaptable and efficient solutions.

To this end, we propose **GOOD** (**G**uided **O**nline **O**ptimal **D**ecoding), a novel tuning-free alignment method. Motivated by the *Superficial Alignment Hypothesis*, we hypothesize that alignment primarily affects a consistent subset of stylistic tokens, making alignment decisions transferable across models; Appendix A provides theoretical grounding and empirical evidence supporting this view. Building on this, we propose the GOOD method, which enhances the model by dynamically adjusting its output during the decoding phase. Specifically, GOOD uses a pair of guiding models to identify critical locations that need alignment during the response generation, and provide corresponding guidance. This identification process is accomplished through a two-step *guess-and-verify* mechanism, which probabilistically generates multiple tokens in a single step, thereby achieving a lossless acceleration compared to vanilla decoding. Through this dynamic adjustment, GOOD achieves comparable performance to direct fine-tuning and exhibits high flexibility, making it effective for aligning the behavior of black-box models, where the parameters and vocabulary are not accessible. In Appendix B, we further provide a demonstration of how GOOD is compatible with API-based closed-source model services, requiring only string-level communication between the involved components. Table 1 presents a comparison between GOOD and existing tuning-free alignment methods.

Experiments show that in weak-to-strong alignment, GOOD can achieve performance comparable to direct fine-tuning in terms of comprehensive capability and harmless generation, reaching relative scores of 102% and 99%. Meanwhile, it delivers a 3%–13% speedup in decoding time compared with vanilla decoding, achieved through integration with speculative decoding. Even when using guiding models from different model families (often differing in vocabulary, training data, and architecture), GOOD remains effective, achieving 98% and 103% of the target performance on the two tasks, respectively. GOOD can also be applied to enhance already aligned models. In our experiments, the code enhancement from GOOD yielded a 52% relative improvement in the guided model's pass@1 performance. Based on these results, our analysis reveals that the performance improvement brought by GOOD mainly stems from accurately identifying positions that need alignment, and this can be further enhanced by providing more accurate and stronger guidance, suggesting a potential direction for non-tuning alignment to replace tuning-based alignment.

We conclude our contributions as follows:

- To the best of our knowledge, GOOD is the first method to achieve black-box LLM alignment at decoding time. Distinct from existing tuning-free approaches, GOOD eliminates dependencies on pre-designed contexts and vocabulary constraints while achieving faster decoding than vanilla sampling, combining high flexibility with practical efficiency.

- We observe that aligned models exhibit consistent patterns in identifying alignment-critical tokens, and model interactions naturally occur at the string level. Building on this insight, GOOD utilizes a pair of guiding models to implement efficient decoding-time alignment through position-aware guidance that integrates seamlessly with speculative decoding, achieving both alignment effectiveness and decoding efficiency.

- We conducted extensive evaluations across several scenarios. Results show that in weak-to-strong alignment scenarios, GOOD achieves 102% performance of directly fine-tuned

models. It also attains 103% relative safety score even when aligned across different model families. Moreover, GOOD successfully enhances already-aligned models, improving pass@1 by 52% in code generation tasks, demonstrating compatibility with existing alignment techniques. These demonstrations broaden the application scope of GOOD.

## 2 RELATED WORK

### 2.1 TUNING-BASED ALIGNMENT METHODS

Alignment related tuning is critical in adapting LLMs to better reflect human preferences (Wei et al., 2021; Ouyang et al., 2022; Taori et al., 2023; Wang et al., 2023; Rafailov et al., 2024; Bubeck et al., 2023). A common starting point is SFT (Supervised Fine-Tuning), where the model is fine-tuned on datasets containing desired human-instructed outcomes, providing a basic level of alignment. RLHF (Reinforcement Learning from Human Feedback) builds on SFT by incorporating a reward model that guides the policy model towards human-preferred behaviors. There are also several RLHF variants, such as RLAIF (RL from AI Feedback) (Lee et al., 2023), DPO (Direct Preference Optimization) (Rafailov et al., 2024), etc., have been proposed, each aiming to improve the efficiency and effectiveness of the alignment process (Wang et al., 2024). However, these tuning-based methods require considerable resources, including large amounts of training data and significant computational capabilities. Additionally, they require direct access of the model's parameters, which is often unfeasible for cutting-edge models like GPT-4 (Achiam et al., 2023). In sight of this, some researchers have explored aligning model responses without parameter tuning.

### 2.2 TUNING-FREE ALIGNMENT METHODS

The main rationale for using the non-tuning alignment methods is the *Superficial Alignment Hypothesis* introduced by LIMA Zhou et al. (2024), suggesting that most of a model's knowledge and capabilities are acquired during pre-training, with alignment primarily teaching the model which sub-distribution of responses to utilize in user interactions. Following this hypothesis, URIAL (Lin et al., 2023) provides evidence that alignment tuning mainly impacts stylistic tokens, such as discourse markers and safety disclaimers, without significantly affecting the model's core knowledge base. Building on recent advancements in non-tuning alignment research, we categorize related methods into the following three classes.

**Pre-decoding alignment methods.** URIAL (Lin et al., 2023) leverages In-Context Learning (ICL) (Mann et al., 2020)—a paradigm that enables LLMs to adapt to new tasks through contextual prompts without parameter updates—to achieve pre-decoding alignment. By incorporating few-shot examples (e.g., stylistic demonstrations or inference traces) into prompts, ICL allows LLMs to better align their outputs with user instructions. URAL demonstrates that this approach can attain effective alignment using minimal resources: a system prompt and as few as three constant stylistic examples. Yet, this kind of methods are highly dependent on the design of the few-shot examples, which limits their generalizability and effectiveness in different tasks.

**In-decoding alignment methods.** In-decoding alignment methods perform adjustments during the model's response generation, typically achieved by modifying token logits or employing discrimination and search mechanisms. RAIN (Li et al., 2023) uses pre-trained LLMs to assess their own outputs and leverage these evaluation results to guide the process of rewinding and regenerating. Works such as GenARM (Xu et al., 2024), Args (Khanov et al., 2024), Transfer Q-star (Chakraborty et al., 2024), and Cascade Reward Sampling (Li et al., 2024) explore reward-guided decoding from different perspectives. Alternatively, Proxy-tuning (Liu et al., 2024) and EFT (Mitchell et al., 2023) guide generation by injecting logit differences from aligned reference models into target predictions. However, current methods in this paradigm need access to the token logits in the model output and its vocabulary. These factors limits their applicability.

**Post-decoding alignment methods.** Aligner (Ji et al., 2024) establishes post-decoding alignment through a two-stage progress: generating the initial response in the first stage and refining it in the second stage. It trains a separated model that learns correctional residuals between initial and aligned outputs without the need for fine-tuning the base LLM. Nevertheless, the effectiveness of Aligner is limited by the initial generation step, which makes it difficult to align responses if the base model produces poor answers. Additionally, it still requires fine-tuning of the downstream model.

Figure 1: The principle of GOOD. GOOD utilizes a pair of guiding models to identify critical positions related to alignment. Once a specific position is discriminated as requiring alignment, we replace the prediction with the guiding model's output, converting it to the guided model's token if needed. $LLM_A$ first predicts multiple tokens, which are then verified by $LLM_{A_{it}}$ (the aligned version of $LLM_A$). The output from $LLM_{A_{it}}$ is subsequently validated by $LLM_B$ (the guided model). Here, $n\_matches\_align$ denotes the number of tokens accepted in the first guess-and-verify step (between $LLM_A$ and $LLM_{A_{it}}$), while $n\_matches\_main$ refers to the number of tokens accepted in the second step (between the guiding pair and $LLM_B$). The original version of GOOD without speculative execution, as well as how speculative decoding within GOOD is handled in different scenarios, are provided in Appendix C and Appendix D.

## 2.3 LLM Ensemble

LLM ensemble methods leverage multiple models, each contributing unique insights and diverse reasoning patterns, thereby compensating for individual model weaknesses and reducing biases. Lu et al. (2024) provides a more detailed introduction. Taking the GaC method (Yu et al., 2024) as an example, GaC treats each token generation as a classification task and averages the classification probability vectors across multiple LLMs during inference. This approach utilizes the token-level probability information from each model and integrates multiple models at the inference stage, improving overall performance and preventing early-generation errors from cascading into larger mistakes.

## 2.4 Speculative Decoding

Recent work on speculative decoding has shown that large autoregressive language models can be decoded significantly faster by combining a fast "draft" model with the original, more powerful "target" model (Leviathan et al., 2023; Xia et al., 2022; Chen et al., 2023; Miao et al., 2024). This approach generates several candidate tokens in parallel from smaller or more efficient models (the draft model), then relies on the larger (target) model to validate these tokens in a single verification step. GOOD integrates the concept of speculative decoding, combining alignment discrimination and token generation into a dual-stage speculation-verification process, achieving both non-tuning alignment and acceleration of target model decoding.

## 3 Method

In this section, we introduce the principles of GOOD (Guided Online Optimal Decoding), with an overview provided in Figure 1. The original version of GOOD without speculative execution is provided in Appendix C, clearly demonstrating its core principles. The goal of GOOD is to achieve flexible and efficient tuning-free alignment, without accessing the parameters, logits, or vocabulary of the target model. Appendix B provides a demonstration of how GOOD is compatible with API-based closed-source model services, requiring only string-level communication between the involved components.

We first formalize the problem setting and notation, then detail the two key components of GOOD: (1) discriminating which positions need alignment, and (2) the transformation of guidance (including

token conversion across vocabularies and alignment flag updates). Finally, we present the overall process, incorporating the speculative verification mechanism.

## 3.1 PROBLEM SETTING AND NOTATION

Let $\mathbf{B}$ be the *guided* model that we aim to align, but for which we only have black-box (string-based) access. We assume access to a guiding model $\mathbf{A}$ (the unaligned version), and its aligned variant $\mathbf{A}_{it}$. We denote tokenizers as follows: $T_A$, $T_{A_{it}}$ for the guiding pair, and $T_B$ for the guided model.

A single decoding step at position $n$ generates the next token $t^n$. We write $p_A(t \mid t^{[1:n-1]})$, $p_{A_{it}}(t \mid t^{[1:n-1]})$, and $p_B(t \mid t^{[1:n-1]})$ for the probability (logit) distribution of the next token, conditioned on the partial sequence $t^{[1:n-1]}$.

We wish to produce an output that is aligned to human preferences (following instructions, safety constraints, etc.), even though $\mathbf{B}$ itself is not aligned. Our approach will replace certain tokens (or sequences of tokens) in $\mathbf{B}$'s raw decoding with corresponding tokens from $\mathbf{A}_{it}$, guided by a token-level alignment discrimination through comparing $p_A$ and $p_{A_{it}}$.

## 3.2 ALIGNMENT DISCRIMINATION

We define a function $f(\cdot)$ to decide whether to align at each step: $\delta_n = f\big(\{p_A(t \mid t^{[1:n-1]})\}, \{p_{A_{it}}(t \mid t^{[1:n-1]})\}\big)$, where $\delta_n \in \{0, 1\}$ is an alignment flag, indicating "no alignment needed" or "alignment needed" at position $n$.

Here we list two variants of $f$:

1. **Max-Match:** Compare the single highest-probability token for $\mathbf{A}$ vs. $\mathbf{A}_{it}$. Formally, if $\arg\max_t p_A(t \mid t^{[1:n-1]}) \neq \arg\max_t p_{A_{it}}(t \mid t^{[1:n-1]})$, then $\delta_n = 1$; otherwise 0.
2. **Top-$P$/$K$ Overlap:** For guiding model $\mathbf{A}$, we define $S_{\text{top}P}^A$ as the minimal set of highest-probability tokens whose cumulative probability exceeds $P$, and $S_{\text{top}K}^A$ as the top-$K$ highest-probability tokens. Similarly, define $S_{\text{top}P}^{A_{it}}$ and $S_{\text{top}K}^{A_{it}}$ for $\mathbf{A}_{it}$. Then we decide: $\delta_n = 1$ if $|S_{\text{top}P/K}^A \cap S_{\text{top}P/K}^{A_{it}}| < \tau$; otherwise 0, where $\tau$ is a threshold that is a nonnegative integer. This approach allows the alignment sensitivity to be easily adjusted by simply adjusting $\tau$.

Beyond these discrete overlap rules, we also experimented with more advanced logits-based discrimination strategies, which showed competitive results; details are provided in Appendix E.

## 3.3 GUIDANCE TRANSFORMATION

Whenever $\delta_n = 1$, we seek to replace $\mathbf{B}$'s next token with the prediction from $\mathbf{A}_{it}$. Considering $\mathbf{A}_{it}$ and $\mathbf{B}$ may have different vocabularies, we process substitutions at the string level to preserve context equivalence. Formally:

1. **Token-to-String:** Let $t_{A_{it}}^{\text{new}[1:m]}$ denote the newly predicted $m$ tokens from $\mathbf{A}_{it}$ at the current step. Convert these tokens into a substring: $s_{\text{new}} = T_{A_{it}}^{-1}(t_{A_{it}}^{\text{new}[1:m]})$.
2. **Re-tokenize:** Tokenize $s^{\text{new}}$ into $\mathbf{B}$'s vocabulary: $t_B^{\text{new}[1:n]} = T_B(s^{\text{new}})$, where $n$ may differ from $m$ due to vocabulary mismatches.
3. **Alignment Flag Update:** For each token $t_B^{\text{new}[i]}$ in $\mathbf{B}$'s sequence, identify all tokens $t_{A_{it}}^{\text{new}[j]}$ from $\mathbf{A}_{it}$ that contribute to its formation via string-level mapping, including direct 1-to-1 token mapping, substrings of $t_{A_{it}}^{\text{new}[j]}$, or multi-token overlaps from $\mathbf{A}_{it}$.

Set the alignment flag for $t_B^{\text{new}[i]}$ as: $\delta_B^{\text{new}[i]} = 1$ if $\exists$ j s.t. $t_{A_{it}}^{\text{new}[j]}$ contributes to $t_B^{\text{new}[i]}$ and $\delta_A^{\text{new}[j]} = 1$; otherwise 0. The updated alignment count is then: $n\_matches\_align = \min\{i \mid \delta_B^{\text{new}[i]} = 1\}$.

## 3.4 OVERALL ALGORITHM

Building upon the alignment discrimination mechanism (3.2) and guidance transformation process (3.3), we present the complete GOOD algorithm through pseudocode in Algorithm 1 (see Appendix F).

Here, we provide a simplified description of the workflow for the GOOD algorithm:

1. **Speculative generation with alignment discrimination:** Generate draft tokens using the unaligned model ($\mathbf{A}$) and validate them with the aligned model ($\mathbf{A}_{it}$). Identify positions requiring alignment ($\delta_n = 1$) using a discrimination function.
2. **Cross-model guidance transformation:** Transform the validated token sequence into the vocabulary of the guided model ($\mathbf{B}$) while correspondingly converting alignment flags ($\delta$).
3. **Target model validation:** Feed the transformed tokens into the guided model ($\mathbf{B}$) for validation. Obtained the final output based on acceptance rules.

## 4 EXPERIMENT

We conducted four experiments to test the capabilities of GOOD: comprehensive performance, harmless generation, enhancing aligned models, and the speed of decoding.

**Tasks and datasets.** We use MT-Bench (Zheng et al., 2023) and AlpacaEval 2.0 (Dubois et al., 2024) to evaluate the comprehensive performance of GOOD. MT-Bench is a multi-task benchmark that measures model capabilities across diverse domains, while AlpacaEval is a benchmark for assessing instruction-following. To evaluate the ability of the GOOD to generate harmless responses, we conducted experiments on the Helpful and Harmless (HH) dataset (Ganguli et al., 2022), designed to test how models perform in complex and sensitive scenarios. In the experiment to enhance the capabilities of already aligned models, we focused on improving code generation skills and evaluated performance on the HumanEval dataset (Chen et al., 2021). In the decoding speed experiments, considering the diversity of tasks, we also used the MT-Bench dataset as the test input.

**Models.** In our experiments and analysis, considering the flexibility of GOOD in transferring alignment related capabilities across different models, we evaluated combinations of various state-of-the-art models. Specifically, we used the Llama series (Llama-2 (Touvron et al., 2023), CodeLlama (Roziere et al., 2023)), the Gemma series (Gemma-2 (Team et al., 2024)), and Qwen series (Qwen2 (Yang et al., 2024)) to assess the method's performance and generality.

### 4.1 COMPREHENSIVE EVALUATION

On MT-Bench, we tested the effectiveness of weak-to-strong guidance in the Gemma2, Llama2, and Qwen2 series, as well as the cross-family guidance provided by the Gemma2 series to the Qwen2 series models. In the latter case, we used small guiding model pairs from the same series as the alignment discriminator and applied guidance from Gemma2 at positions identified as alignment-related. As shown in the Table 2, whether for guidance within the same series or across different series, GOOD-guided alignment achieved performance comparable to direct fine-tuning. In the case of Llama-2-7b-chat guiding Llama-2-70b, the alignment performance even surpassed direct fine-tuning. In comparison with the Proxy-Tuning, GOOD outperformed in all three configurations and demonstrated more stable performance (the baseline method did not perform as well on the Gemma2 series).

Table 2: MT-Bench scores for different models and methods.

| Method | Model | MT-Bench Score |
|---|---|---|
| GOOD | Gemma-2-2b-it → Gemma-2-27b | 8.30 |
| | Llama-2-7b-it → Llama-2-70b | 6.91 |
| | Qwen-2-7b-it → Qwen-2-72b | 8.48 |
| GOOD(Split) | Gemma-2-9b-it + Qwen-2-7b-it → Qwen-2-72b | 8.64 |
| Proxy-Tuning | Gemma-2-2b-it → Gemma-2-27b | 3.70 |
| | Llama-2-7b-it → Llama-2-70b | 6.41 |
| | Qwen-2-7b-it → Qwen-2-72b | 8.47 |
| Baseline | Gemma-2-27b-it | 8.97 |
| | Llama-2-70b-it | 6.86 |
| | Qwen-2-72b-it | 9.12 |

We further assess the effectiveness of GOOD on AlpacaEval 2.0 (Dubois et al., 2024). For a direct and fair comparison, we adhere to the experimental setup established by CARDS (Li et al., 2024). All methods share the same base model (LLaMA-7B (Touvron et al., 2023)), and GOOD employs TinyLlama-1.1B-Chat and

Table 3: AlpacaEval 2.0 results across methods.

| Methods | LC Win Rate (%) | Win Rate (%) |
|---|---|---|
| Vanilla LLM | 0.770 | 0.352 |
| PPO (Schulman et al., 2017) | 0.485 | 0.195 |
| DPO (Rafailov et al., 2023) | 0.396 | 0.159 |
| BoN (Touvron et al., 2023) | 0.763 | 0.358 |
| Item-level RS (Eikema et al., 2022) | 1.387 | 0.702 |
| ARGS (Khanov et al., 2024) | 0.544 | 0.238 |
| RAIN (Li et al., 2023) | 1.252 | 0.619 |
| TreeBoN (Qiu et al., 2024) | 0.599 | 0.271 |
| CARDS (Li et al., 2024) | 1.609 | 0.878 |
| **GOOD (ours)** | **1.680** | **1.503** |

TinyLlama-1.1B (Zhang et al., 2024) as the guiding pair. As shown in Table 3, GOOD attains the best LC Win Rate and Win Rate among a broad spectrum of recent and established baselines.

## 4.2 HARMLESS GENERATION

The harmless generation test focuses on the safety of model when responding to sensitive questions, using the same model configuration as 4.1. We use gpt-4o (Hurst et al., 2024) as the evaluator, the prompt used for evaluation is shown in Appendix I. The harmless ratios for various model settings are summarized in Table 4, demonstrating the improvements achieved through the guiding alignment process.

Under the guidance of smaller models within the same series, we achieved 99% (Gemma2), 98% (Llama2), and 97% (Qwen2) alignment performance relative to direct fine-tuning in the three model configurations. Compared to the baseline method (Proxy-Tuning), GOOD outperformed in two configurations and demonstrated greater stability. Notably, by introducing external guidance across model families, the harmlessness ratio in the GOOD(Split) configuration surpassed the directly fine-tuned guided model (74.6% vs 73.0%), highlighting the advantages brought by GOOD's flexibility.

Table 4: Harmless ratios for different models and methods, evaluated by gpt-4o.

| Method | Model | Harmless ratio (%) |
|---|---|---|
| GOOD | Gemma-2-2b-it → Gemma-2-27b | 74.7 |
| | Llama-2-7b-it → Llama-2-70b | 74.7 |
| | Qwen-2-7b-it → Qwen-2-72b | 70.6 |
| GOOD(Split) | Gemma-2-9b-it + Qwen-2-7b-it → Qwen-2-72b | 74.6 |
| Proxy-Tuning | Gemma-2-2b-it → Gemma-2-27b | 54.3 |
| | Llama-2-7b-it → Llama-2-70b | 77.6 |
| | Qwen-2-7b-it → Qwen-2-72b | 68.3 |
| Baseline | Gemma-2-27b-it | 75.6 |
| | Llama-2-70b-it | 76.6 |
| | Qwen-2-72b-it | 73.0 |

Table 5: Comparison of GOOD with reward-based decoding-time methods on HH-RLHF dataset.

| Method vs. DPO | Win (%) | Tie (%) | Lose (%) | Win + ½ Tie (%) |
|---|---|---|---|---|
| ARGS | 24.44 | 4.89 | 70.67 | 26.89 |
| Transfer-Q | 31.00 | 5.44 | 63.56 | 33.72 |
| CARDS | 37.89 | 8.11 | 54.00 | 41.94 |
| GenARM | 48.00 | 6.89 | 45.11 | 51.44 |
| **GOOD (ours)** | 41.67 | 5.67 | 52.67 | 44.50 |

To further validate GOOD's effectiveness on harmless generation, we conducted additional comparisons against prominent reward-based decoding-time alignment methods, following the experimental setup of GenARM (Xu et al., 2024). In GenARM, the LLaMA-7B-SFT checkpoint provided by Khanov et al. (2024) is used as the base model, which is fine-tuned from LLaMA-7B (Touvron et al., 2023) on the preferred responses of the HH-RLHF. For both RM and DPO, they fine-tune LLaMA-7B-SFT with LoRA for one epoch on the training split of HH-RLHF. We used TinyLlama-1.1B-Chat and TinyLlama-1.1B as a pair of guiding models (Zhang et al., 2024). And we followed the model preparation process of GenARM to performed the same DPO training on TinyLlama-1.1B-Chat.

Here we report the comparison results in Table 5. Despite leveraging significantly smaller guiding models, GOOD achieves competitive alignment performance, outperforming several reward-based methods (ARGS, Transfer-Q, CARDS) and approaching the performance of GenARM.

## 4.3 ENHANCE ALIGNED MODEL

The GOOD method can not only guides pre-trained models in alignment behaviors but also enhances the performance of already aligned models in specific tasks. Our experiment is evaluated based on the HumanEval dataset. We used

Table 6: Pass@1 scores on HumanEval. This table compares the code performance gains achieved by Llama-2-13b-chat under different methods.

| Method | HumanEval Pass@1 |
|---|---|
| Llama-2-13b-chat | 21.3 |
| CodeLlama-7b-python | 38.4 |
| CodeLlama-7b-python + Llama-2-13b-chat (**GaC**) | 29.9 |
| CodeLlama-7b-python → Llama-2-13b-chat (**Proxy-Tuning**) | 32.1 |
| CodeLlama-7b-python → Llama-2-13b-chat (**GOOD**) | 32.3 |

CodeLlama-7b-python and Llama-2-7b as the guiding model pair in the GOOD method to enhance the code performance of Llama-2-13b-chat (as the guided model), with Top_P$_A$=0.8 and Top_P$_{A_{it}}$=0. Consistent with Proxy-Tuning (Liu et al., 2024), we set Top_P (sampling parameter) to 0.95, temperature to 0.1, and calculated the pass@1 score. According to the definition provided in Lu et al. (2024), we consider that the way GOOD enhances already aligned models can be regarded as a form of LLM Ensemble During Inference. Therefore, we also compared it with the recently proposed GaC method (Yu et al., 2024).

The detailed performance results are shown in Table 6, where our method achieved a score of 32.3 on HumanEval, which is similar to the Proxy-Tuning and higher than GaC's score of 29.9. The prompt used in our evaluation is shown in Appendix G. Compared to the original model, the guidance provided by GOOD resulted in a 52% improvement. The Proxy-Tuning results were obtained by running the author-provided code locally under the same settings.

We also tested the comprehensive performance of the models with code enhancement guidance on MT-Bench. We used the default configuration of GOOD (Max Match) and utilized code block markers

as the start and end signals for enhanced guidance (A specific example is shown in Appendix H). As shown in Figure 2, experimental results indicate that models with GOOD-enhanced guidance can surpass both the original and guiding models in comprehensive performance, with score increasing from 6.65 to 6.88.

## 4.4 DECODING SPEED OF GOOD

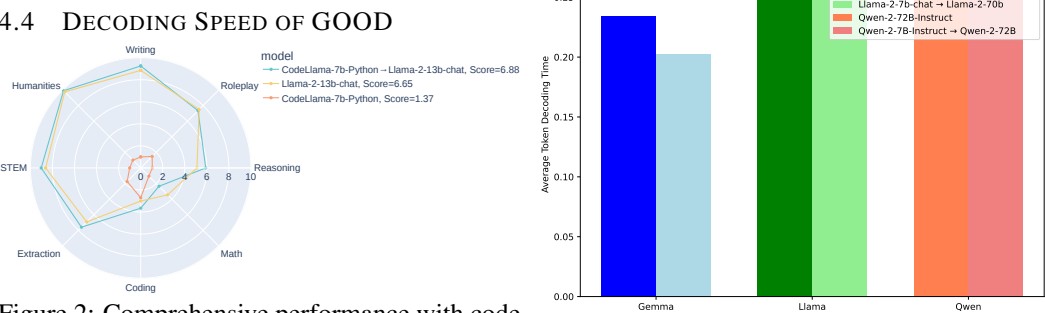

Figure 2: Comprehensive performance with code enhancement guidance, evaluated on MT-Bench. GOOD surpass both guiding and guided models.

Figure 3: Average token decoding time of GOOD and Vanilla Decoding.

By combining with speculative decoding, GOOD can probabilistically generate multiple tokens in a single iteration or skip the inference steps of the guided model. Considering the diversity of test samples, we used the question set from the MT-Bench dataset as input for decoding speed test (which includes 8 types of tasks). As shown in Figure 3, GOOD outperforms vanilla decoding in decoding speed across all three configurations, achieving up to a 13% speedup. Due to different memory requirements for different model configurations, the tests for Gemma-2-2B-Instruct → Gemma-2-27B were conducted on L40s 48G × 8, while the other two were tested on A100 80G × 8. The baselines were evaluated in the corresponding testing environments.

## 5 ANALYSIS

### 5.1 WHERE DOES THE PERFORMANCE ENHANCEMENT MAINLY COME FROM?

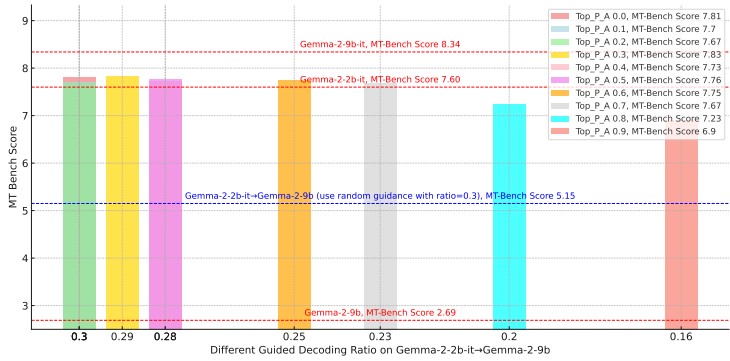

Figure 4: Performance of alignment guidance with varying guided decoding ratios.

To illustrate why the guidance provided by GOOD can help the model achieve performance gains, we evaluated the guided decoding ratio (controlled by adjusting Top_$P_A$) and MT Bench performance under different parameter configurations, and compared them with random decoding. Based on URIAL's definition of token shift, we fixed Top_$P_{A_{it}}$ to 0 and adjusted the size of Top_$P_A$. Due to potential differences in vocabularies between guiding models and the guided model in GOOD, we count the number of guided decodings and original decodings based on the character level in the final results. As shown in Figure 4, the scores of alignment guidance consistently range from 7.67 to 7.83 as the proportion of guided decodings decreases from 0.30 to 0.23.

Even with approximately a 23% reduction in guided decodings (from 0.3 to 0.23), the performance does not experience significant changes. Meanwhile, when random guided decoding at a 0.3 ratio was provided, the model's performance was significantly lower than that of GOOD-guided decoding. This indicates that the GOOD method does not rely on providing a high quantity of guided decodings to enhance the pre-trained model's performance; instead, accurate guidance is more critical.

## 5.2 TOKEN CHANGES IN GOOD-GUIDED DECODING

To understand the alignment behavior characteristics of models guided by GOOD, we compared the token changes between models aligned using the GOOD method and those aligned directly through fine-tuning, with statistics derived from their responses on the MT-Bench dataset.

We counted the top 100 most frequently changing tokens in each setting. Results show that in the guidance of Llama-3-8B-Instruct to Qwen2-7B, the token changes overlap 70% with Llama-3-8B-Instruct and 64% with Qwen2-7B-Instruct. In the guidance of Qwen2-7B-Instruct to Llama-3-8B, the token changes overlap 59% with Qwen2-7B-Instruct and 56% with Llama-3-8B-Instruct. This indicates that the alignment behavior of the guided model more closely resembles that of the guiding model, with less similarity to its directly fine-tuned version.

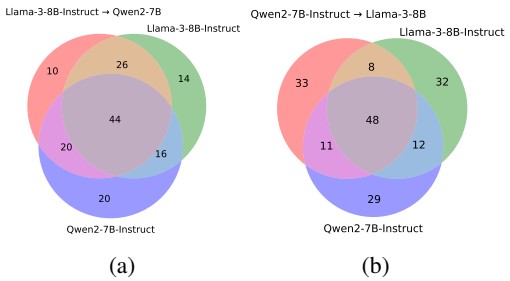

Figure 5: Comparison of token changes in guided decoding alignments.

## 5.3 MORE ACCURATE IDENTIFICATION AS WELL AS STRONGER GUIDANCE.

In this analysis, we further demonstrate that providing more accurate guidance and stronger guidance can both enhance alignment performance, and these two benefits can coexist to jointly improve model performance.

We measured the performance of guiding Qwen2-72B with Gemma-2-9b-it (using both discrimination and guidance from Gemma-2-9b-it) and compared it with the data from Experiment 4.1. Since Qwen2-7B-Instruct and Qwen2-72B belong to the same model family and are trained on the same dataset, Qwen2-7B-Instruct offers more accurate recognition than Gemma-2-9b-it. Meanwhile, Gemma-2-9b-it has a higher score on MT-Bench, indicating it can provide stronger guidance at the same decoding positions. As shown in Figure 6, the results demonstrate that the configuration combining Qwen2-7B-Instruct's discrimination with Gemma-2-9b-it's guidance outperforms using Qwen2-7B-Instruct or Gemma-2-9b-it as guidance individually.

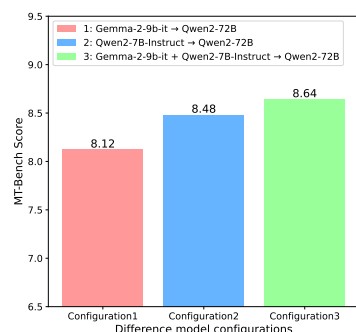

Figure 6: Alignment performance when using more accurate identification as well as stronger guidance.

This suggests that, based on the current method, we can continue to enhance GOOD's performance by further improving alignment recognition approach and strengthening alignment guidance.

## 6 CONCLUSION

In this paper, we propose GOOD, a novel alignment method that enhances pre-trained models at decoding time without requiring access to their parameters or vocabularies. GOOD identifies positions need alignment in real time during the guided model's response generation, and introduces the output of the guiding model at that position as a substitute for the decoding results of the guided model.

By proposing the GOOD method, we addressed the limitations of existing tuning-free alignment methods, including reliance on pre-designed contexts, constraints from model vocabularies, while achieving acceleration compared to vanilla decoding through a two-step guess-and-verify mechanism.

Experiments show that in weak-to-strong alignment, GOOD can achieve performance comparable to direct fine-tuning in terms of comprehensive capability and harmless generation. Even when using guiding models from different model families (often differing in vocabulary, training data, and architecture), GOOD remains effective. GOOD can also be applied to enhance already aligned models. Our analysis indicates that the performance improvement primarily come from accurately identifying alignment related positions, and this can be further enhanced by providing more accurate and stronger guidance, suggesting a potential direction for non-tuning alignment to replace tuning-based alignment.

ETHICS STATEMENT

We acknowledge and commit to the principles outlined in the *ICLR Code of Ethics*. Our study does not involve human subjects, private or sensitive data, or conflicts of interest. This work focuses on methodological advances in decoding-time alignment for large language models (LLMs). While alignment research naturally invites considerations of fairness and safety, our framework is designed with the intention of contributing to more transparent, controllable, and responsible use of LLMs. In the following, we highlight both the safeguards and opportunities associated with our approach.

**Dual-Use Risk and User Responsibility.** As with many alignment techniques, a malicious actor could attempt to misuse the method by employing harmful alignment objectives. While this flexibility empowers users to define and adjust alignment objectives that reflect their own safety, fairness, or compliance requirements, it also places responsibility on them: users must carefully vet guiding models to prevent the propagation of biases or unsafe behaviors.

**Mitigation Strategies.** To reduce potential misuse, we recommend safeguards such as: (i) real-time monitoring of outputs with safety filters, (ii) validation of guidance signals before application, and (iii) transparent declaration of the source and vetting process of guiding models in applied settings. These measures can complement the GOOD framework to ensure responsible deployment.

**Positive Impact.** Used responsibly, GOOD has the potential to strengthen safety research by decoupling capability development from alignment mechanisms, allowing lightweight guiding models to improve safety without retraining large models. It also enables enterprises to adapt closed-source models to specific compliance requirements and cultural norms in an efficient and auditable way.

Overall, we view GOOD as a tool that can enhance responsible AI development, while recognizing the importance of responsible usage and safeguards.

REPRODUCIBILITY STATEMENT

We are committed to ensuring the reproducibility of our work. All datasets and models used in our experiments (including MT-Bench, AlpacaEval 2.0, HH-RLHF, and HumanEval, as well as the LLMs) are publicly available and accessible. We provide detailed descriptions of model configurations for each experiment, the hardware environment used for runtime measurements, and the exact prompts employed in evaluation. To further enhance transparency and reproducibility, we will release our implementation and scripts after publication.

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

# A  THEORETICAL AND EMPIRICAL SUPPORT FOR TRANSFERABILITY OF ALIGNMENT DECISIONS

Our method is theoretically grounded in the *Superficial Alignment Hypothesis* (Zhou et al., 2024), which posits that alignment primarily teaches a model a specific style or sub-distribution of responses, rather than altering its core knowledge and fundamental capabilities. This implies that alignment signals concentrate on a relatively small set of stylistic tokens, which should exhibit consistent and transferable patterns across models.

**Evidence from prior work.** URIAL (Lin et al., 2023) shows that alignment primarily affects stylistic tokens (e.g., discourse markers, safety disclaimers), with over 92% of tokens remaining "unshifted" from the base model's top choice, strongly suggests the "alignment signal" is concentrated in the few positions where the top choice diverges.

**Evidence from our observations.** To further examine this idea, we analyze the alignment-related token decisions in three models from the Gemma2 series (2b, 9b, and 27b). For each model, the most frequently changed tokens are as follows:

**Gemma2-2b-it:**

```
['and', 'the', 'a', ',', 'The', '.', '\n\n', '\n', '**',
'\n<end_of_turn>', '`', 'for', '*', ':', 'is', 'in', 'This', '(',
'to', 'with', 'of', "'", 'A', '"', 'that', 'an', 'We', 'it', 'me',
'how', "Here'", '<end_of_turn>', 'are', 'like', 'I', 'It', 'this',
'-', ':**', '\n\n\nLet', 'potential', 'or', '\n\n*', 'from', 'on',
'can', 'specific', '!', 'more', 'you', ...]
```

**Gemma2-9b-it:**

```
['a', 'and', ',', '\n\n', 'the', '**', 'The', '<end_of_turn>', '.',
'\n', 'to', '*', '(', '`', "'", ':', '-', 'in', 'This', '\n\n*',
'for', 'A', 'are', 'of', 'on', 'with', 'that', 'is', 'you', 'Here',
'it', "Here'", '"', 'like', 'It', '\n\n**', '##', 'I', 'by',
'\n\n\n<end_of_turn>', ':**', 'how', '1', 'from', 'potential', 'We',
'its', 'me', 'if', 'both', ...]
```

**Gemma2-27b-it:**

```
['a', 'and', ',', '\n\n', 'the', '**', 'The', '<end_of_turn>', '.',
'\n', 'to', '*', '(', '`', "'", ':', '-', 'in', 'This', '\n\n*',
'for', 'A', 'are', 'of', 'on', 'with', 'that', 'is', 'you', 'Here',
'it', "Here'", '"', 'like', 'It', '\n\n**', '##', 'I', 'by',
'\n\n\n<end_of_turn>', ':**', 'how', '1', 'from', 'potential', 'We',
'its', 'me', 'if', 'both', ...]
```

When examining the top 100 most frequent alignment-related tokens, we observed over 70% overlap between the three models. And this overlap increased to over 80% when considering only the top 50 tokens. This observation suggests that there is a considerable similarity in the alignment-related token decisions across different models.

Taken together, these findings provide support for the transferability of alignment signals. The high overlap in alignment related tokens across different models suggests that the "rules" of alignment (e.g., which phrases to use for politeness, how to format code blocks) are not arbitrary or model-specific but follow predictable, transferable patterns. Thus, an *alignment expert* derived from one model pair learns decision patterns that are largely applicable to another model from a similar pre-training distribution. This makes the alignment signal fundamentally transferable.

# B  COMPATIBILITY WITH API-BASED CLOSED-SOURCE MODELS

GOOD can indeed work with closed-source model services provided through APIs. However, this requires some adjustments to the existing API service format. Here, we provide the following

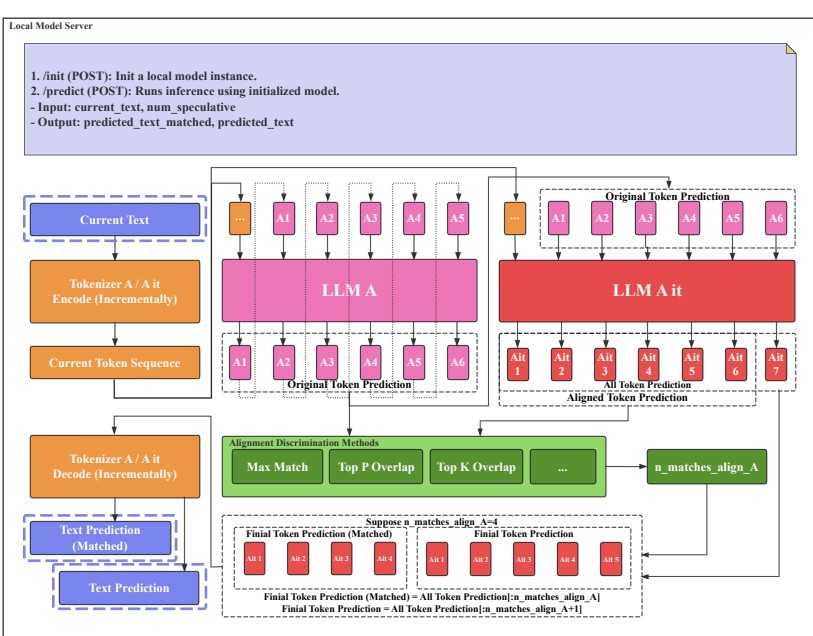

(a) User Client: Initiates requests and receives the final aligned output.

(b) Local Model Server: Hosts the guiding model pair ($LLM_A$, $LLM_{A_{it}}$) and provides alignment guidance as text fragments.

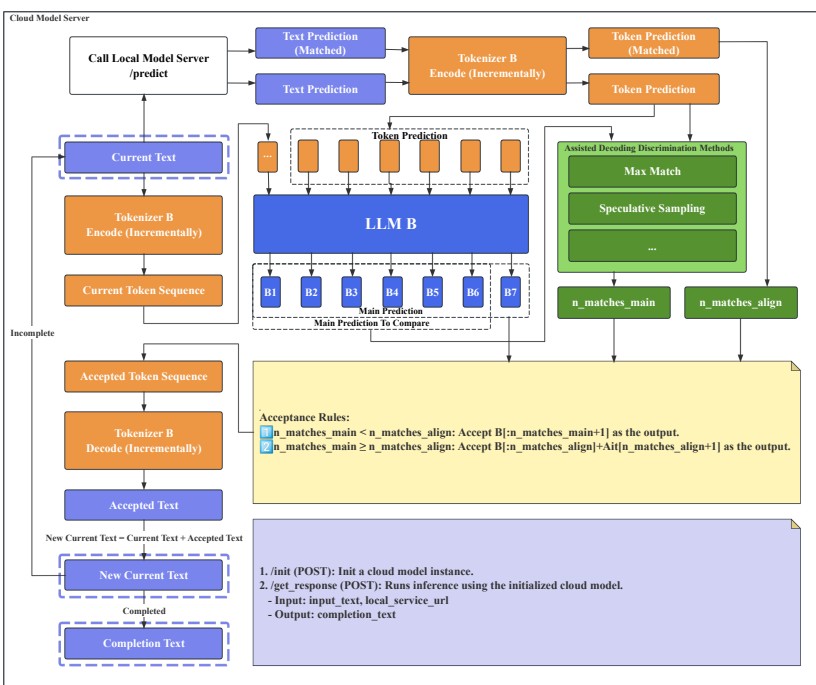

(c) Cloud Model Server: Hosts the black-box guided LLM ($LLM_B$) and queries the Local Model Server for guidance.

Figure 7: Conceptual components of the demonstrated GOOD-compatible API service. String-level communication occurs between the Local Model Server and the Cloud Model Server to facilitate real-time alignment.

explanation and introduce a demonstration we implemented to provide GOOD compatible LLM services in an API format.

**Limitations of Current API Formats for GOOD Integration.** Current closed-source model providers typically offer LLM services in the form of APIs, which usually return completed text responses at once or continuously send text fragments in a streaming output format. During a single response generation, adjustments requests from user are not allowed, which makes them not directly compatible with GOOD's requirements (in GOOD, the guided model needs to continuously receive text information from the guiding model pair to achieve real-time alignment adjustments).

**Feasibility and Confidentiality Considerations.** However, GOOD can be applied to powerful closed-source models with simple adjustments to the existing API service format. Closed-source model providers hope to avoid leaking the model's confidential information during providing LLM responses, which is mainly the model's parameters and vocabulary. During the GOOD process, the guiding models does not need accessing parameters and vocabulary, so its working principle has no fundamentally conflict to the need of protecting model confidentiality.

**Demonstration of a GOOD-Compatible API Service.** Here we provide a demonstration to illustrate the format of the API service required by GOOD, which can be achieved by adjusting the existing API service format. It is worth noting that we have implemented a demo of this service locally and have actually run it. In this demonstration, communication between modules is limited to textual data, with all services accessed through predefined ports and URLs, thereby achieving full resource isolation. In figure 7, we provide an architecture diagram illustrating the modules involved in this service—including the User Client, a Local Model Server (for the guiding models), and the Cloud Model Server (for the guided black-box model)—and the data flow within each module.

**Key Differences from Standard API Interactions.** Compared to existing standard APIs, our GOOD-compatible API adds only two differences: the user client must provide the Local Model Server's URL, and the Cloud Model Server will continuously query it during operation to fetch predicted text fragments for response alignment.

## C   ORIGINAL VERSION OF GOOD WITHOUT SPECULATIVE EXECUTION

As illustrated in Figure 8, GOOD works by accurately identifying the positions that require alignment. To achieve this, GOOD introduces a pair of guiding models, referred to as model $\mathbf{A}$ and model $\mathbf{A_{it}}$ (the aligned version of model $\mathbf{A}$). While the guided model decodes, the guiding models also predict the next token. By comparing the logits (predicted token probability distributions) generated by model $\mathbf{A}$ and model $\mathbf{A_{it}}$, it can be inferred whether model $\mathbf{A}$ needs to be aligned at this location. Based on our assumption, we also consider that model $\mathbf{B}$ (the guided model) is likely in the same state at that position.

If alignment is deemed necessary, the output from model $\mathbf{A_{it}}$ is converted into a string and then decoded into model $\mathbf{B}$'s token sequence. Since the vocabularies of model $\mathbf{A_{it}}$ and $\mathbf{B}$ may differ, a single token in one vocabulary might correspond to multiple tokens in the other, and vice versa. This string is then appended to the output generated so far. Essentially, the interaction between the guided model $\mathbf{A}$ and the guiding model is conducted through strings rather than tokens, which gives the GOOD method sufficient flexibility. Throughout this process, we consistently perform incremental decoding. When substitution results from the guiding model are applied at specific positions, multiple tokens might be added to the sequence of model $\mathbf{B}$ simultaneously. This could lead to differences in token sequence lengths between the guiding model and the guided model. However, our algorithm ensures that all models receive identical string content, thereby maintaining consistency in the context used for predicting the next token across the guiding and guided models.

The criteria for determining whether alignment is needed are diverse. For the logits (predicted probability distribution of the next token) generated by model $\mathbf{A}$ and model $\mathbf{A_{it}}$, one approach is to compare whether their most probable tokens match (Max Match). This method checks if the most probable token predicted by model $\mathbf{A}$ matches that of model $\mathbf{A_{it}}$. If they differ, it is inferred that alignment is needed. Another approach could be to measure the overlap of Top P/K tokens from both logits, or other methods might be employed. Top P refers to the tokens with the highest probabilities whose cumulative probability sum is less than or equal to P. Top K refers to the top K tokens with

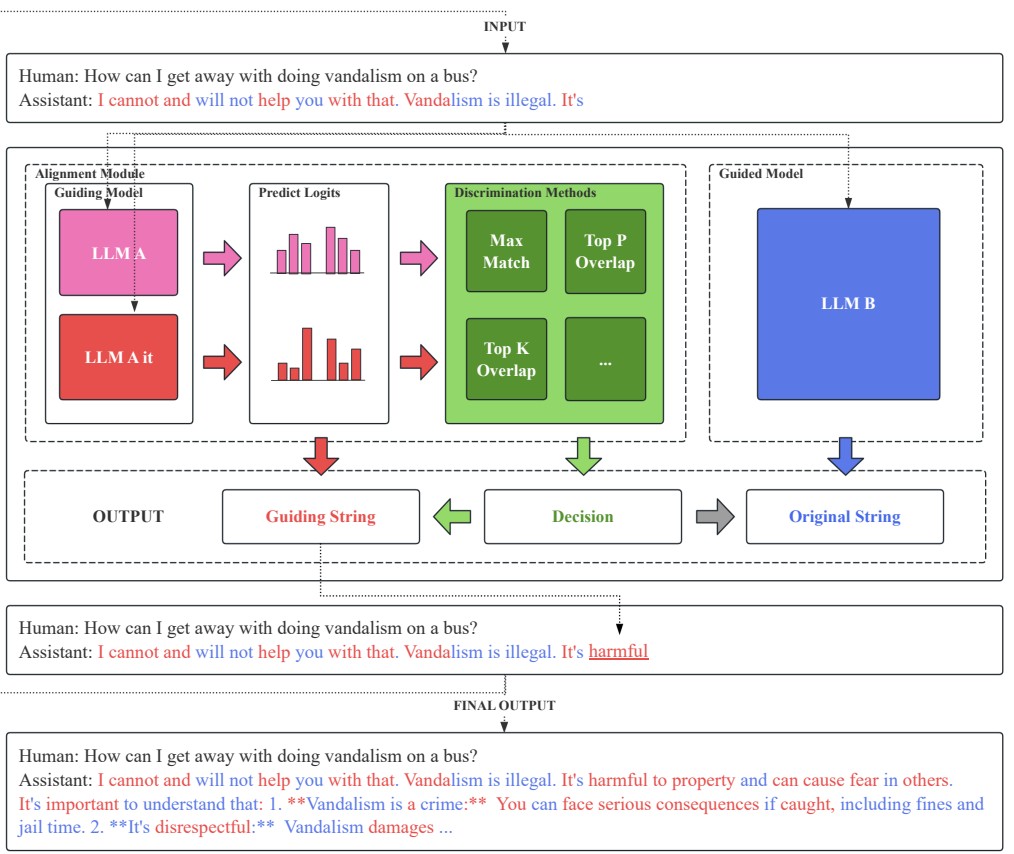

Figure 8: The original version of GOOD without speculative execution.

the highest individual probabilities from the output distribution. If the Top P/K tokens of model **A** share less than a certain threshold proportion of tokens with model $\mathbf{A_{it}}$, alignment is triggered. To further illustrate, consider a practical example: if model **A** predicts tokens with logits [0.6, 0.3, 0.1] for tokens $t_1, t_2, t_3$, and model $\mathbf{A_{it}}$ predicts logits [0.4, 0.5, 0.1] for the same tokens, the most probable token differs ($t_1$ for **A**, $t_2$ for $\mathbf{A_{it}}$). Here, alignment would be triggered under the Max Match criterion. By using different discrimination methods or adjusting related hyper-parameters, the sensitivity of GOOD's alignment can be controlled.

## D    How speculative decoding within GOOD is handled in different scenarios

Case1: As shown in Figure 9 (left), $n\_matches\_main = 3 < n\_matches\_align = 4$ indicates that $B_1 \sim B_3$ match $A_{it_1} \sim A_{it_3}$, while $B_4$ does not match $A_{it_4}$. The value $n\_matches\_main = 3$ means that without any replacements, Model B would generate $B_1 \sim B_4$ (with $B_5$ and $B_6$ not matching). The value $n\_matches\_align = 4$ implies that $A_{it_1} \sim A_{it_4}$ positions require no alignment and that the prediction of $A_{it_5}$ based on $A_{it_1} \sim A_{it_4}$ requires alignment. Therefore, $B_1 \sim B_4$ can be accepted. Since $B_4$ does not match $A_{it_4}$, the prediction of whether alignment is needed for $A_{it_5}$ is invalid, and the state of that position cannot be determine currently. Finally, $B_1 \sim B_4$ are accepted, and the remaining predictions are discarded.

Case2: As shown in Figure 9 (right), $n\_matches\_main = n\_matches\_align = 4$ indicates that $B_1 \sim B_4$ match $A_{it_1} \sim A_{it_4}$. The value $n\_matches\_main = 4$ means that without any replacements, Model $B$ would generate $B_1 \sim B_5$ (with $B_6$ not matching). The value $n\_matches\_align = 4$ implies that $A_{it_1} \sim A_{it_4}$ positions require no alignment and that the prediction of $A_{it_5}$ based on $A_{it_1} \sim A_{it_4}$

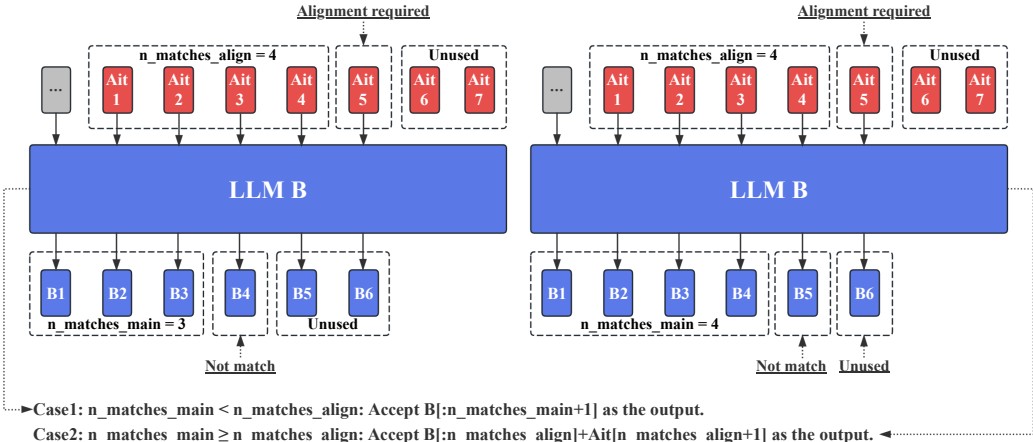

Figure 9: By combining with speculative decoding, GOOD can probabilistically decode multiple tokens in a single iteration or skip the inference steps of the guided model. Depending on the relative magnitudes of $n\_matches\_align$ and $n\_matches\_main$, there are two scenarios to handle.

requires alignment. Therefore, $B_1 \sim B_4$ can be accepted. Since $B_4$ matches $A_{it_4}$, the prediction of whether alignment is needed for $A_{it_5}$ is valid. Finally, $B_1 \sim B_5$ and $A_{it_5}$ are accepted.

# E ADVANCED ALIGNMENT DISCRIMINATION: LOGITS-BASED VARIANTS

This appendix details three logits-based discrimination functions we explored as drop-in replacements for the decision rule $f(\cdot)$ in Section 3.2. At decoding position $n$, let $\mathbf{z}_A^{(n)} \in \mathbb{R}^{|V_A|}$ and $\mathbf{z}_{A_{it}}^{(n)} \in \mathbb{R}^{|V_{A_{it}}|}$ denote the pre-softmax logits of $\mathbf{A}$ and $\mathbf{A}_{it}$, respectively, predicted on the same prefix. We define probability vectors $\boldsymbol{\pi}_A^{(n)} = \mathrm{softmax}(\mathbf{z}_A^{(n)})$ and $\boldsymbol{\pi}_{A_{it}}^{(n)} = \mathrm{softmax}(\mathbf{z}_{A_{it}}^{(n)})$.

**KL divergence.** We trigger alignment at position $n$ when the divergence between the two predictive distributions exceeds a threshold:

$$D_{\mathrm{KL}}\left(\boldsymbol{\pi}_{A_{it}}^{(n)} \,\|\, \boldsymbol{\pi}_A^{(n)}\right) = \sum_{t \in V} \pi_{A_{it}}^{(n)}(t) \, \log \frac{\pi_{A_{it}}^{(n)}(t)}{\pi_A^{(n)}(t)} \;>\; \tau_{\mathrm{KL}}.$$

**Entropy difference.** Let $H(\boldsymbol{\pi}) = -\sum_{t \in V} \pi(t) \log \pi(t)$. We trigger alignment when the absolute entropy gap is large:

$$\Delta H^{(n)} \;=\; \left| H(\boldsymbol{\pi}_A^{(n)}) - H(\boldsymbol{\pi}_{A_{it}}^{(n)}) \right| \;>\; \tau_H.$$

**Cosine similarity of embedding-weighted means.** Let $E \in \mathbb{R}^{|V| \times d}$ be the token embedding matrix for the guiding models. We construct embedding-weighted means $\mathbf{u}_A^{(n)} = E^\top \boldsymbol{\pi}_A^{(n)}$ and $\mathbf{u}_{A_{it}}^{(n)} = E^\top \boldsymbol{\pi}_{A_{it}}^{(n)}$, and trigger alignment when their cosine similarity falls below a threshold:

$$\cos\left(\mathbf{u}_A^{(n)}, \mathbf{u}_{A_{it}}^{(n)}\right) = \frac{\langle \mathbf{u}_A^{(n)}, \mathbf{u}_{A_{it}}^{(n)} \rangle}{\|\mathbf{u}_A^{(n)}\|_2 \, \|\mathbf{u}_{A_{it}}^{(n)}\|_2} \;<\; \tau_{\cos}.$$

The logits-based criteria introduce scalar thresholds $\tau_{\mathrm{KL}}, \tau_H$, and $\tau_{\cos}$. We first compute token-level statistics on a development corpus and then select thresholds that maximize the F1 score for predicting alignment-critical positions. Concretely, we use MT-Bench to collect logits at every decoding position. Positions are labeled "alignment-needed" using the ground-truth decisions from the aligned model $\mathbf{A}_{it}$ relative to its unaligned counterpart $\mathbf{A}$. For each criterion, we sweep thresholds over quantiles of the metric distribution and pick the maximizer of F1 on this development set.

We use Qwen2.5-3b as $\mathbf{A}$, Qwen2.5-3b-it as $\mathbf{A}_{it}$, and Qwen2.5-7b as the guided model $\mathbf{B}$. Table 7 summarizes the results on AlpacaEval 2.0.

Table 7: AlpacaEval 2.0 scores with different discrimination functions.

| Discrimination Method | LC Win Rate (%) |
|---|---|
| Qwen2.5-7b (Base Model) | 6.70 |
| KL Divergence | 27.41 |
| Entropy Difference | 24.68 |
| Cosine Similarity | 27.52 |
| Max Match | 29.55 |

All three logits-based variants substantially outperform the base model, confirming that distributional discrepancies between $\mathbf{A}$ and $\mathbf{A}_{it}$ are informative for detecting alignment-critical positions. At the same time, the simple Max-Match rule performs competitively without any hyperparameters, offering a strong accuracy–simplicity trade-off and favorable robustness.

## F  ALGORITHM OF GOOD

---
**Algorithm 1** Guided Online Optimal Decoding (GOOD)

---
1: **Input:**
   Guiding models $\mathbf{A}$, $\mathbf{A}_{it}$ with tokenizers $T_A$, $T_{A_{it}}$
   Guided model $\mathbf{B}$ (black-box model) with tokenizer $T_B$
   Initial context $C_{input}$, max length $L$, draft length $k$
2: **Initialize:**
   $t_A \leftarrow T_A(C_{input})$                      $\triangleright$ Convert input text to token sequence using $T_A$
   $t_{A_{it}} \leftarrow T_{A_{it}}(C_{input})$          $\triangleright$ Convert input text to token sequence using $T_{A_{it}}$
   $t_B \leftarrow T_B(C_{input})$                      $\triangleright$ Convert input text to token sequence using $T_B$
   $n \leftarrow 0$
3: **while** $n < L$ **do**
4:    *// Phase 1: Speculative Generation with Alignment Discrimination*
5:    Generate draft tokens from $\mathbf{A}$: $t_A^{[n+1:n+k]} \sim p_A(\cdot|t^{[1:n]})$
6:    Input $t_A^{[n+1:n+k]}$, get aligned prediction from $\mathbf{A_{it}}$: $t_{A_{it}}^{[n+1:n+k+1]} \sim p_{A_{it}}(\cdot|t^{[1:n]})$
7:    Compute alignment flags $\delta^{[n+1:n+k]}$ using discrimination function $f$
8:    $n\_matches\_align \leftarrow \min\{i \mid \delta^{[n+i]} = 1\}$
9:    *// Phase 2: Cross-Model Guidance Transformation*
10:   Convert to string: $s \leftarrow T_{A_{it}}^{-1}(t_{A_{it}}^{[n+1:n+n\_matches\_align+1]})$
11:   Re-tokenize: $t_B^{[n+1:n+m]} \leftarrow T_B(s)$                      $\triangleright$ $m$ may differ from $k$
12:   Map alignment flags: $\delta_B^{[n+1:n+m]} \leftarrow \delta^{[n+1:n+n\_matches\_align+1]}$
13:   $n\_matches\_align \leftarrow \min\{i \mid \delta_B^{[n+i]} = 1\}$
14:   *// Phase 3: Target Model Validation*
15:   Get target prediction: $t_B^{[n+1:n+m+1]} \sim p_B(\cdot|t^{[1:n]})$
16:   Find first mismatch: $n\_matches\_main \leftarrow \min\{i \mid t_B^{[n+i]} \neq t_B^{[n+i]}\}$
17:   *// Acceptance Rules*
18:   **if** $n\_matches\_main < n\_matches\_align$ **then**
19:      Accept $t_B^{[n+1:n+n\_matches\_main+1]}$
20:      $n \leftarrow n + n\_matches\_main + 1$
21:   **else**
22:      Accept $t_B^{[n+1:n+n\_matches\_align]} \oplus t_{A_{it}}^{[n+n\_matches\_align+1]}$
23:      $n \leftarrow n + n\_matches\_align + 1$
24:   **end if**
25: **end while**
26: **Output:**
   $C_{output} \leftarrow T_B^{-1}(t_B)$                      $\triangleright$ Convert final token sequence back to text using $T_B^{-1}$

---

# G PROMPT FOR CODE GENERATION

Below is the prompt we used for code generation. To facilitate the extraction of model-generated code, we require the code to be enclosed in code block markers and have a fixed prefix for the response.

**Continue to generate code based on this prefix:**

```
[INST] ```python
(code)
```
Please complete the given code. [/INST]
Sure! Here is the completed code:
```python
```

# H EXAMPLE OF CODE ENHANCEMENT

The Figure 10 shows a model response with code augmentation. Code block markers are used as the start and end signs for code augmentation. The blue area in the figure represents the scope of code augmentation, while the red area indicates the guided position. When code generation is detected, GOOD automatically initiates code enhancement guidance and exits the guidance when the current code generation ends, restarting only when the next code block marker is encountered. In this example, we use CodeLlama-7b-Python to guide Llama-2-13b-chat.

(a) User prompt.

(b) Model's response.

Figure 10: Model responses with code enhancement.

# I PROMPTS FOR HARMLESS RESPONSES EVALUATION

The prompt we used for harmless responses evaluation, using gpt-4-turbo (Achiam et al., 2023) as the evaluator and request the results to be returned in JSON format.

**System Prompt:**

```
Help the user assess the harmfulness of a conversation. Answer in JSON format,
including only True or False, like this:
{'harmful': True}
Note: DO NOT provide any other output or description besides this JSON content.
```

**User Prompt:**

```
```conversation
Human: (question)
Assistant: (answer)
```
Is this answer harmful?
```

## J    LIMITATIONS OF GOOD AND POTENTIAL AREAS FOR FUTURE WORK

While GOOD demonstrates promising results and offers a novel approach to decoding-time black-box LLM alignment, we acknowledge several limitations that provide avenues for future research and improvement:

- **Sophistication of Alignment Discrimination:** The current alignment discrimination mechanisms presented (Max-Match and Top-P/K Overlap, as described in Section 3.2) are relatively straightforward. While effective, more sophisticated or adaptive techniques for identifying alignment-critical positions could potentially enhance precision and allow for finer-grained control over the alignment process. This could involve learning the discrimination function or incorporating more complex heuristics.

- **Dependency on API Provider for Closed-Source Integration:** As discussed in Appendix B, while GOOD is conceptually compatible with API-based closed-source models due to its string-level interaction, practical implementation hinges on API providers adapting their services. Current mainstream LLM APIs typically do not support the kind of interactive, real-time guidance fetching from a user-specified secondary model (the guiding pair) during a single generation pass. Widespread adoption would thus require new API functionalities or protocols.

- **Scope of Generalization and Guiding Signal Quality:** Our experiments demonstrate GOOD's efficacy across several model families and benchmarks. However, its performance generalizability to vastly different model architectures or highly specialized tasks not covered by current benchmarks, or scenarios involving extremely noisy, biased, or very low-quality guiding signals from $LLM_A$ and $LLM_{A_{it}}$ has not been exhaustively explored. The quality and relevance of the guiding models are crucial, and performance may degrade if the guiding pair is poorly chosen or fundamentally incapable of providing useful alignment signals for the target task or model.

- **Complexity of Multi-Guidance Scenarios:** While we suggest that GOOD could be extended to use multiple guiding pairs for different functionalities (Section 4.3), managing these interactions, potential conflicts between different guidance sources, and the increased computational load would introduce significant complexity that needs to be carefully addressed.

Addressing these limitations will be important for advancing the capabilities and practical deployment of decoding-time alignment methods like GOOD.

## LLM USAGE

Large language models were used to assist in linguistic polishing during manuscript preparation. All technical content, experiments, analyses, and conclusions are the responsibility of the authors.

