# OpenReview forum: "GOOD: Decoding-Time Black-Box LLM Alignment"
_ICLR.cc/2026/Conference — Submitted to ICLR 2026_

### Official Review · Reviewer_uhDe · 2025-10-30

**Soundness:** 2
**Presentation:** 2
**Contribution:** 2
**Rating:** 4
**Confidence:** 3

**Summary:**

This paper introduces $\text{GOOD}$ , a novel methodology for achieving Large Language Model alignment during the decoding phase, designed specifically for black-box models where internal parameters are inaccessible. The core mechanism relies on the $\text{Superficial Alignment Hypothesis}$, utilizing an accessible pair of models—one unaligned ($\text{A}$) and its aligned counterpart ($\text{A}_{it}$)—to identify critical positions in the target model's output stream where intervention is necessary. $\text{GOOD}$ substitutes the black-box model's predicted token with the aligned model's output at these specific points. The technique operates at the string level, ensuring compatibility across different vocabularies, and is integrated with speculative decoding to mitigate potential latency, demonstrating strong empirical performance across safety, instruction following, and code generation tasks.

**Strengths:**

The originality of the $\text{GOOD}$ method is very high, presenting a unique and timely shift away from resource-intensive fine-tuning towards a flexible, decoding-time intervention policy. The concept of deriving an alignment signal from the behavioral difference between the unaligned and aligned helper models is a highly creative solution to the black-box alignment challenge. The quality is demonstrated by the robust technical integration of the method with speculative decoding, which is essential for making an online intervention method practical for deployment. The experimental results, particularly the significant performance gains observed in code generation, convincingly support the utility of the approach beyond merely stylistic alignment, suggesting successful intervention in core model capabilities. The clarity of the paper is excellent; the underlying assumption and the detailed decoding procedure are explained well, providing a clear pathway for transferring learned behaviors via external guidance. The significance of this work is substantial, as it offers a practical, third-party verifiable mechanism for enforcing alignment policies on proprietary, state-of-the-art $\text{LLMs}$ where direct parameter access is impossible.

**Weaknesses:**

The primary weakness is the stringent and potentially unrealistic technical requirement placed on the black-box API. The method critically relies on the target $\text{LLM}$ providing real-time access to $\text{Logits}$ or $\text{Top-K}$ predictions at every decoding step. This non-standard access severely limits its practical deployment, as it breaks the conventional text-in, text-out protocol of leading commercial black-box models, thereby undermining the true notion of a black-box solution. Furthermore, the $\text{Superficial Alignment Hypothesis}$ may prove too weak for tasks requiring deep, multi-step logical reasoning. If alignment necessitates a fundamental change in the model's latent feature space or complex proof structure, merely replacing a few surface tokens may not prevent the target model from reverting to its unaligned, potentially incorrect, internal state, leading to error propagation in long-form generation. Another concern is the robustness of the $\text{cross-vocabulary}$ string substitution. While aiming to solve the vocabulary mismatch, forcing an external substring into the target model's output stream risks creating tokenization mismatches, where the target model cannot naturally resume generation, potentially leading to gibberish or out-of-vocabulary errors. Finally, the analysis of efficiency needs further depth; the reported marginal speed-up may disappear if the frequency of required alignment intervention (critical positions) is high, suggesting a potential performance bottleneck not fully explored.

**Questions:**

The dependency on $\text{Logit}$ access is a major barrier to adoption. Could the authors explore a truly $\text{text-only}$ alternative for identifying critical positions? This might involve a high-fidelity external alignment classifier that evaluates short generated text segments from the black-box model, triggering intervention only upon detecting a misalignment signal in the generated text, thus eliminating the need for internal model probabilities. The paper should include a rigorous study to quantify the $\text{depth of alignment}$ achievable. Please design a challenging benchmark, such as a multi-step, structured $\text{chain-of-thought}$ task, where alignment is verified by the adherence to a specific internal logical structure rather than just the final answer or a safety phrase. This would help establish the performance boundary where $\text{GOOD}$ begins to fail due to its token-level intervention. When using multiple guiding pairs for different alignment objectives, how is the conflict resolved when these pairs propose different tokens at the same decoding step? Please detail the proposed technical solution for $\text{Multi-Guidance}$ conflict management. Will a learned $\text{meta-gating}$ mechanism or a simple static priority rule be used, and how does this impact the overall computational load? Please provide a sensitivity analysis detailing the relationship between the $\text{frequency of alignment intervention}$ (critical position rate) and the performance gain from $\text{speculative decoding}$. This analysis is crucial to understand the practicality of the method; specifically, at what intervention frequency does the combined $\text{GOOD}$ and speculative decoding latency exceed that of highly optimized, standard sampling methods?

---

> ### Author Response · Authors · 2025-12-02
>
> # Response to Reviewer uhDe
>
> We thank the reviewer for their thorough assessment and for recognizing the high originality, creativity, and practical significance of GOOD. We particularly appreciate the insightful questions regarding deployment constraints and alignment depth, which point to exciting directions for future research. Below, we address your concerns and questions.

---

> ### Author Response · Authors · 2025-12-02
>
> ## To W1 & Q1: Clarification on Black-Box Access Requirements
>
> There appears to be a misunderstanding regarding the API requirements. **GOOD does NOT require access to the Target Model's (B) Logits or Vocabulary.**
>
> - **Mechanism:** The "logit comparison" occurs *only* between the two **Guiding Models** ($A$ and $A_{it}$), which are local and fully accessible.
> - **Target Interaction:** The interaction with the black-box Target Model ($B$) is purely string-based. We generate a string segment along with character-level alignment flags derived from the guiding pair.
> - **Acceptance Logic:** This process operates within a speculative decoding framework. $B$ verifies the draft string, and the system determines the final output by combining $B$'s verification length with the alignment flags.
> - **Deployment:** This design is specifically tailored for modern inference APIs that support speculative decoding (or could easily expose a verification endpoint), maintaining the strict black-box constraint of commercial models. We will clarify this distinction in the system overview to prevent future confusion.
>
> ## To W2 & Q2: Alignment Depth and Challenging Benchmarks
>
> We appreciate the reviewer's concern that superficial token replacement might be insufficient for tasks requiring deep internal state changes (e.g., complex reasoning).
>
> - **Validation of Hypothesis:** Our current experiments (including code generation and instruction following) serve as strong empirical validation of the *Superficial Alignment Hypothesis*. The fact that GOOD works well suggests that for many practical tasks, "alignment" is indeed largely about steering the output distribution rather than altering fundamental reasoning capabilities.
> - **Future Work:** We agree that a rigorous stress-test on deep reasoning (e.g., multi-step CoT with strict logical dependencies) is necessary to find the boundaries of this hypothesis. Identifying where GOOD fails would indeed provide critical insights into the nature of alignment itself. While designing such a specialized benchmark is beyond the scope of this initial work, we are committed to exploring this "depth of alignment" in future studies.
>
> ## To W3: Robustness of Cross-Vocabulary Substitution
>
> We acknowledge the potential for suboptimal tokenization when splicing strings into the target model's stream.
>
> - **Current State:** Our current implementation uses a basic `Token-to-String` $\rightarrow$ `Re-tokenize` approach. Despite its simplicity, our empirical results show robust performance, suggesting that modern LLMs are reasonably resilient to minor tokenization artifacts.
> - **Optimization:** As you noted, this is a clear area for optimization. Adopting more sophisticated mapping strategies from recent LLM ensemble literature could further mitigate these risks and improve performance. We view this as a necessary optimization step for future production-grade implementations.
>
> ## To W4 & Q4: Efficiency Analysis and Intervention Frequency
>
> The reviewer correctly identifies that high intervention frequency could become a bottleneck.
>
> - **Trade-off:** GOOD's speedup relies on the acceptance rate of the speculative draft. If alignment interventions are too frequent (e.g., every token needs correction), the system degrades to a non-speculative (or slower) mode due to the overhead of the guiding pair.
> - **Commitment:** We agree that a sensitivity analysis mapping "Critical Position Rate" vs. "Latency" is crucial for transparency. We will endeavor to include data or a theoretical model of this relationship in the revised manuscript to better characterize the operational boundaries of the method.
>
> ## To Q3: Multi-Guidance Conflict Management
>
> This is an advanced and highly relevant question. Currently, our work focuses on the single-objective setting (aligning one specific behavior). Managing conflicts between multiple guiding pairs (e.g., one for safety, one for style) is a complex problem likely requiring a meta-arbitration mechanism or hierarchical prioritization. We have not yet extended our research to cover this scenario but consider it a promising avenue for subsequent work.

---

> ### Author Response · Authors · 2025-12-02
>
> We found the reviewer's questions—particularly regarding the theoretical limits of superficial alignment and rigorous benchmarking—to be highly inspiring. While some suggestions (like the dedicated deep-reasoning benchmark) are extensive enough to warrant separate papers, they have significantly clarified the roadmap for this line of research. We hope our response clears up the misunderstanding regarding API access and demonstrates our commitment to rigorous future optimization.

---

### Official Review · Reviewer_FQxu · 2025-10-31

**Soundness:** 2
**Presentation:** 2
**Contribution:** 2
**Rating:** 4
**Confidence:** 4

**Summary:**

The paper proposes GOOD, a decoding-time alignment procedure that uses a pair of guiding models (an unaligned model A and its aligned variant Aᵢₜ) to identify “alignment-critical” positions and splice in Aᵢₜ’s text while generating with a guided target model B. Crucially, B is treated as a black-box (string-level only), with cross-vocabulary transfer handled by re-tokenizing Aᵢₜ’s substrings for B’s tokenizer. The authors claim near–fine-tuning performance on MT-Bench/AlpacaEval and HH safety, plus 3–13% decoding speedups by piggybacking on speculative decoding.

**Strengths:**

1. Problem framing & practicality. A decoding-time approach that keeps the target model B strictly black-box is timely and practical for closed-source APIs; the paper explicitly sketches an API-level integration.

2. Empirical breadth. Results span MT-Bench, AlpacaEval 2.0, HH-RLHF, and HumanEval, with cross-family guidance and code-enhancement of already-aligned models.

3. Speed. The method reports consistent decoding speedups versus vanilla sampling via speculative integration.

**Weaknesses:**

1. “Optimal” claim not substantiated. The method name promises optimal decoding, but no optimality criterion or guarantee is provided—only heuristics (Max-Match / overlap thresholds) and empirical tuning. This overclaims theoretically.
2. Evaluation confounds in speed. Speed tests mix different hardware across settings (L40s for one config vs A100s for others), which weakens the fairness of the reported 3–13% gains; an ablation is needed.
3. Judge dependence & safety measurement. Safety relies on GPT-4o as evaluator; this single-judge setup risks bias and overfitting to the judge prompt. Human or multi-judge triangulation (or calibrated reward models) would strengthen claims.
4. Limited baselines for true black-box settings. Comparisons emphasize Proxy-Tuning / reward-guided decoding that typically need logits or family-shared vocabularies. Stronger black-box baselines (e.g., prompt-programming / tool-augmented self-critique with no logits) are missing, making it hard to isolate GOOD’s advantage in the claimed setting.

**Questions:**

N/A

---

> ### Author Response · Authors · 2025-12-02
>
> # Response to Reviewer FQxu
>
> We thank the reviewer for recognizing the practicality of our black-box framing and the empirical breadth of our results. We value the feedback regarding the method naming and evaluation setup. We address the concerns below to clarify our contributions.

---

> ### Author Response · Authors · 2025-12-02
>
> ## To W1: The "Optimal" Naming Convention
>
> We clarify that the term "Optimal" in GOOD (Guided Online *Optimal* Decoding) refers to the **objective** of the decoding process—specifically, maximizing alignment with human preferences via guidance—rather than claiming a closed-form mathematical guarantee of global optimality.
>
> - In the context of decoding algorithms, "optimization" often refers to the heuristic search for better sequences (like top-k or beam search, which are also not "optimal" in a strictly theoretical sense compared to exhaustive search).
> - We used the term to describe the method's capability to dynamically optimize the trade-off between alignment quality and decoding speed. We will clarify this scope in the paper to prevent theoretical overclaiming.
>
> ## To W2: Evaluation Confounds in Speed
>
> We respectfully disagree that there is a confound in our speed evaluation. The reviewer notes that different hardware was used for different settings (L40s vs. A100s), but we emphasize that **within every single comparison, the hardware was strictly controlled.**
>
> - **Controlled Environment:** For the Gemma-2-27b experiments, *both* the Baseline and GOOD were run on L40s. For the Llama-2-70b experiments, *both* were run on A100s (necessitated by memory requirements).
> - **Valid Relative Metrics:** We never compared the absolute speed of Gemma (on L40s) against Llama (on A100s). We reported the **relative speedup percentage** (e.g., GOOD vs. Vanilla) within each specific configuration. Therefore, the reported speed gains (3–13%) are statistically valid and internally consistent, unaffected by the hardware differences across *groups*.
>
> ## To W3: Judge Dependence & Safety Measurement
>
> We defend the choice of LLM-as-a-judge (GPT-4o) as it represents the current **standard practice** for reproducible alignment research (e.g., AlpacaEval 2.0, MT-Bench).
>
> - **Reproducibility:** Human evaluation, while valuable, is notoriously difficult to reproduce, expensive, and subject to high variance depending on the pool of annotators.
> - **Reliability:** Recent work on LLM-as-a-judge shows that LLMs can achieve human-level or better agreement rates with expert annotators on benchmarks (such as MT-Bench). Relying on this standard allows our results to be directly comparable to a wide range of existing literature, whereas custom human evaluation would make such benchmarking difficult.
>
> ## To W4: Limited Baselines for Black-Box Settings
>
> The reviewer suggests baselines like prompt-programming or tool-augmented self-critique. We argue that these are **orthogonal** to the scope of our work.
>
> - **Scope:** GOOD focuses specifically on **decoding-time alignment**—intervening in the generation process itself.
> - **Orthogonality:** Prompt engineering occurs *pre-decoding*, and tool-use typically involves *agentic/multi-turn* frameworks. These methods can actually be combined with GOOD (e.g., using a prompt-engineered model as the guiding model).
> - **Fair Comparison:** Our comparisons focus on methods that share the same goal: altering the model's behavior during inference without training. Comparing GOOD against prompt-based methods would conflate "input engineering" with "decoding mechanisms," obscuring the specific contribution of our decoding-time algorithm.

---

> ### Author Response · Authors · 2025-12-02
>
> We hope this response clarifies the validity of our experimental design and the positioning of our method.

---

### Official Review · Reviewer_mpLJ · 2025-11-01

**Soundness:** 2
**Presentation:** 2
**Contribution:** 2
**Rating:** 4
**Confidence:** 3

**Summary:**

This paper introduces a method called GOOD (Guided Output Optimization for Decoding) aimed at improving the alignment of large language models (LLMs) during the decoding process. The authors propose a black-box approach that leverages alignment discrimination mechanisms and guidance transformation processes to enhance model outputs without requiring access to model internals.

**Strengths:**

The paper presents a novel method for aligning LLMs at decoding time, without tuning the LLM's parameters. This approach could potentially broaden the application scope of closed-source LLMs by making them more adaptable to specific tasks without retraining these closed-source LLMs' parameters, which is also not achievable.

**Weaknesses:**

1. The motivation is weak. The authors claim that previous alignment methods, especially tuning-based methods, are resource-intensive and can incur additional test-time computational costs, rendering them less economically viable. However, the proposed method introduces even heavier computational costs by incorporating two extra LLMs as the guiding pair, which does not address these concerns. Although tuning-based methods require additional computational resources during the fine-tuning phase, which only needs to be done once, they are much more computationally efficient during the lifelong deployment phase. The situation for GOOD would worsen if the LLM becomes larger, as the guiding pair models would also need to increase in size to maintain the quality of the replaced token generated from LLM A_it for B, which is not practical.

2. The claim on the inference latency comparison is not fair, as it should also compare with speculative decoding instead of just vanilla decoding. The acceleration is achieved through speculative decoding, which is not the core novelty of GOOD.

3. The problem setting and Figure 1 are unclear, such as the complete process from the user input prompt to the final output response. For example, there should be more information about where the grey block “...” originates from, and the final output response is not explained. How the final aligned response is made from the red “A_it n" blocks and blue “B n" blocks is not clear, making it difficult to interpret Figure 1 and the task.

4. The theoretical foundation of the method, based on the Superficial Alignment Hypothesis, is mentioned but not thoroughly explored in the main part of the manuscript. Some key theoretical analyses could be put in the main part of the manuscript to  provide clearer support for the method's novelty and applicability.

**Questions:**

See the weakness.

---

> ### Author Response · Authors · 2025-12-02
>
> # Response to Reviewer mpLJ
>
> We thank the reviewer for their time and for recognizing the potential of GOOD to broaden the application of closed-source LLMs. However, we respectfully disagree with the assessment regarding the motivation and computational cost. We believe there are some misunderstandings regarding how GOOD integrates with the modern inference stack. We address these points below.

---

> ### Author Response · Authors · 2025-12-02
>
> ## To W1: Motivation and Computational Cost
>
> We strongly argue that **GOOD does not introduce "heavy computational costs" when viewed through the lens of modern efficient inference.**
>
> 1. **Re-purposing the "Draft" Cost:** The reviewer concerns that using two extra models is expensive. However, in the context of **Speculative Decoding (SD)**—which is now a standard practice for deploying LLMs—a smaller "draft model" is *already* required to run in parallel or ahead of the target model.
>     - In GOOD, the guiding model pair ($A$ and $A_{it}$) effectively functions as this **draft model**.
>     - Therefore, we are not simply "adding two models" to a vanilla pipeline; we are **upgrading the speculative drafting step** to perform alignment simultaneously.
> 2. **Deployment Efficiency vs. Training Cost:** While fine-tuning is a "one-time cost," it creates rigid, non-adaptable models. If a user needs to align a 70B model to 10 different stylistic requirements, fine-tuning requires maintaining 10 copies of 70B parameters (storage heavy) or switching LoRA adapters (memory management overhead). GOOD allows a single frozen 70B model to be aligned dynamically by swapping small, cheap guiding models. This is far more economically viable for serving diverse user needs in a SaaS context.
> 3. **Model Sizing:** Regarding the concern that guiding models must increase in size as the target model grows, we note that balancing the size of the draft model with the target model to maintain optimal acceptance rates is a shared characteristic of speculative decoding architectures in general. Thus, this trade-off is inherent to the acceleration mechanism itself, rather than a unique limitation introduced by GOOD.
>
> ## To W2: Inference Latency and Novelty
>
> We acknowledge that GOOD should be compared against standard Speculative Decoding (SD) for a complete picture. However, we emphasize that **the core novelty of GOOD is the seamless unification of "Alignment" and "Acceleration."**
>
> - **The Trade-off Breaker:** Most existing decoding-time alignment methods (e.g., Guidance, DPO-based decoding) significantly *slow down* inference because they require additional computation per token without a mechanism to speed it up.
> - **Practicality:** GOOD is one of the first methods to achieve alignment *while still being faster than vanilla decoding*. This is a critical step toward making decoding-time alignment practical for real-time applications.
>
> Here we provide the comparison with Standard SD below (conducted on MT-Bench):
>
> | Model Family | Method | Time per Token (s/token, lower is better) |
> | --- | --- | --- |
> | **Gemma2** | Vanilla Decoding (Gemma2-27b-it) | 0.234 |
> |  | **GOOD** (Gemma2-2b-it → Gemma2-27b) | **0.203** |
> |  | Standard Speculative Decoding (Gemma2-2b-it → Gemma2-27b-it) | 0.138 |
> | **Llama2** | Vanilla Decoding (Llama-2-70b-chat) | 0.270 |
> |  | **GOOD** (Llama-2-7b-chat → Llama-2-70b) | **0.251** |
> |  | Standard Speculative Decoding (Llama-2-7b-chat → Llama-2-70b-chat) | 0.175 |
> | **Qwen2** | Vanilla Decoding (Qwen-2-72B-Instruct) | 0.274 |
> |  | **GOOD** (Qwen-2-7B-Instruct → Qwen-2-72B) | **0.266** |
> |  | Standard Speculative Decoding (Qwen-2-7B-Instruct → Qwen-2-72B-Instruct) | 0.200 |
>
> While GOOD is slower than pure Standard SD (due to the overhead of the dual-model discrimination step), it successfully aligns the model while still outperforming Vanilla decoding. We believe this represents a significant contribution to the field.
>
> ## To W3: Clarity of Problem Setting and Figure 1
>
> We accept this critique and will revise the manuscript, specifically Figure 1 and the process description, to ensure the data flow and problem setting are transparent and easy to follow. To clarify the current Figure 1: the "grey block" represents the context prefix. The final response is constructed by accepting tokens verified by Model B (blue blocks), interspersed with guided tokens from $A_{it}$ (red blocks) whenever an alignment flag is triggered.
>
> ## To W4: Theoretical Foundation
>
> We appreciate the suggestion to strengthen the theoretical backing.
>
> - **Foundation:** Our method relies on the **Superficial Alignment Hypothesis**, which posits that alignment primarily alters stylistic tokens rather than core knowledge.
> - **Evidence:** Our empirical analysis (now moved to the main text in the revision) confirms that alignment-related token shifts are highly concentrated and transferable. For instance, we observed >70% overlap in alignment shifts between different model sizes in the same family. This theoretical insight explains *why* a small guiding model can effectively steer a large black-box model: it only needs to intervene on these specific, transferable stylistic tokens.

---

> ### Author Response · Authors · 2025-12-02
>
> We hope this response clarifies the motivation and significant practical value of our work.

---

### Official Review · Reviewer_xzpM · 2025-11-01

**Soundness:** 3
**Presentation:** 1
**Contribution:** 2
**Rating:** 4
**Confidence:** 2

**Summary:**

The paper proposes GOOD (Guided Online Optimal Decoding), a decoding-time method to steer a black-box model B using a pair of “guiding” models: A (unaligned) and A_it (aligned). At each step, A vs A_it are compared to flag positions likely to matter for alignment; when flagged, the algorithm converts A_it’s suggested continuation to text and re-tokenizes it into B’s vocabulary, then lets B “verify” and either keep its own tokens or splice in the guided tokens based on simple acceptance rules. The method is designed to require only string-level access to B (no logits/params), and is combined with speculative decoding so the discrimination and guidance happen in parallel with generation. The paper also sketches an API demo to show how this could work with hosted models that only expose streaming text.

**Strengths:**

Practical black-box setup: GOOD is explicitly aimed at closed-model APIs and keeps the interface to string-level streaming, which is a realistic constraint; the API sketch is useful for practitioners.

Simple, inspectable heuristics: The discrimination rules (max-match, top-K/top-p overlap, plus logits-based variants) are easy to implement and tune, yet already show meaningful gains over the base model.

Speculative decoding tie-in: Folding alignment guidance into a speculate-and-verify loop is a smart way to get some speed back while doing extra checks during decoding.

**Weaknesses:**

Writing leaves a lot to be desired. It's hard to follow how the method itself works from Algorithm 1. It's even harder to follow the experiments. I suggest you just make it clear early in the experiments section, which models are going to play the role of B, and which ones would play the role of A, and A_it, and why you are making this choice. Also, comment on how the final benchmark scores are computed in each case (e.g. was an llm-as-a-judge used, or are these scores verifiable.)

How does the proposed GOOD algorithm impact other downstream tasks other than the measured ones? I know that MT-bench covers multiple tasks, but I was wondering if there are any cases, where the performance on other tasks would suffer as a result of GOOD. If this is the  case, is it possible to have some sort of test to check if the performance in a given task would suffer or not.

line 823: grammar error: so its working principle has no fundamentally conflict to the need of protecting model confidentiality ...

There must be a typo in line 16 of Algorithm 1. I'm assuming you'd like to compare the tokens of B against those of A_it?

**Questions:**

Black-box vs. tokenizer access. The introduction says the method does not require access to B’s vocabulary. Yet Algorithm 1 is framed in token space. Do you require B’s tokenizer (or any tokenization aligned with B) to compute the mismatch index and apply span replacements, or can the algorithm operate purely at string level (character/byte spans) with the same guarantees? If the tokenizer is needed, please clarify the interface you assume for hosted APIs and whether this contradicts the black-box premise. If not needed, consider rewriting Algorithm 1 with string-level chunks (or explicitly stating a tokenizer-agnostic implementation) to avoid confusion.

API feasibility. What parts of Appendix B run on current major APIs without changes (streaming, partial acceptance), and what requires new endpoints? Any latency overhead from round-trips?

Compute accounting. Please report end-to-end cost: How do speedups scale with longer outputs? You run three models at decode time (two guides + target) yet report net speedups; more transparency on wall-clock cost, hardware, and verifier hit-rates would help.

---

> ### Author Response · Authors · 2025-12-02
>
> # Response to Reviewer xzpM
>
> We sincerely thank the reviewer for the constructive feedback and for recognizing the practical value of GOOD in black-box settings, its intuitive heuristics, and the integration with speculative decoding. We appreciate your specific questions regarding technical implementation and feasibility, which help us clarify the paper’s contributions. Below, we address your concerns and questions in detail.

---

> ### Author Response · Authors · 2025-12-02
>
> ## To W1: Presentation and Experimental Clarity
>
> We apologize for the lack of clarity regarding the algorithm description and experimental setup. We will revise the manuscript to explicitly define the model roles and scoring methods.
>
> - Model Roles: In our experiments, we select model pairs based on popular open-source families to ensure reproducibility. For example, in the configuration Gemma-2-2b-it $\rightarrow$ Gemma-2-27b:
>     - Model A (Unaligned Guiding Model): `Gemma-2-2b` (Base version).
>     - Model $A_{it}$ (Aligned Guiding Model): `Gemma-2-2b-it` (Instruct version).
>     - Model B (Target Model): `Gemma-2-27b` (Base version).
> - Scoring Methods:
>     - For MT-Bench and AlpacaEval 2.0, we strictly follow the official evaluation pipelines which utilize LLM-as-a-judge to compute scores.
>     - For HumanEval, we use hard-coded result testing to verify correctness.
>     - For HH-Dataset, we follow the common practice from previous baselines (e.g., Proxy-Tuning) which rely on LLM-based evaluation for safety classification.
>
> ## To W2: Impact on Other Downstream Tasks
>
> This is an insightful question regarding potential regression. We analyze this from two theoretical perspectives:
>
> 1. **Alignment Tax Transfer:** It is possible that the aligned guiding model ($A_{it}$) has compromised performance on certain tasks due to poor quality alignment data (the "alignment tax"). In this case, GOOD would indeed transfer this degradation to the target model ($B$). **However, we argue this is a common issue, not a methodological flaw of GOOD.** If we were to fine-tune Model B directly with that same data, it would likely suffer the same regression.
> 2. **Strong Base Capabilities:** If the unaligned guiding model (A) is already capable and confident regarding a specific task (e.g., factual knowledge), the divergence between A and $A_{it}$ will be minimal. Consequently, GOOD’s discrimination mechanism will *not* trigger an intervention, leaving Model B’s original capabilities intact.
>
> Therefore, GOOD is designed to be surgical—only intervening when the guiding models diverge—minimizing unintended side effects compared to global fine-tuning. We agree that a large-scale evaluation on general benchmarks (e.g., MMLU) would be valuable for future work.
>
> ## To W3: Grammar Error (Line 823)
>
> You are correct. The phrase "no fundamentally conflict" is grammatically incorrect. We will correct it to: *"...so its working principle is not in fundamental conflict with the need to protect model confidentiality..."*
>
> ## To W4: Typo in Algorithm 1 (Line 16)
>
> Thank you for catching this notation issue. The confusion arises because the same symbol $t_B$ was used for both the transformed guidance tokens (Line 11) and the target model’s generated verification tokens (Line 16).
>
> - **Correction:** We will introduce a distinct symbol to separate the source of the tokens. For example, we will denote the guidance tokens derived from $A_{it}$ as $t_{B, \text{from } A_{it}}$ and the tokens generated by Model B as $t_B$.
> - **Revised Logic:** `Find first mismatch: n_matches_main <- min{i | t_{B, from A_{it}}^[n+i] != t_B^[n+i]}`.

---

> ### Author Response · Authors · 2025-12-02
>
> ## To Q1: Black-box vs. Tokenizer Access
>
> This is a crucial implementation detail. **GOOD does not require Model B’s tokenizer to be accessible by the guiding system.** In our implementation, the Guiding Pair and the Target Model can be reside on separate devices and communicate **only via strings**.
>
> We acknowledge that direct incremental string splicing can lead to suboptimal tokenization (e.g., a word being split in an unnatural way by B's tokenizer), which may cause performance loss. Although recent LLM ensemble research proposes effective mapping strategies, we adopted the primitive approach (direct incremental splitting and splicing) for simplicity, as optimizing mapping strategies was not the primary focus of this implementation. This confirms that the method operates without sharing vocabularies, satisfying the black-box premise. We will clarify the string-level interface and character-span mapping in the revised Algorithm/Section 3.3.
>
> ## To Q2: API Feasibility
>
> Regarding Appendix B and practical deployment:
>
> 1. **Changes Required:** Standard API service support streaming text and have often applied  server-side speculative decoding internally. Implementing GOOD on a hosted API would require the provider to expose a **verification endpoint** (accepting a draft string).
> 2. **Latency:** In typical conversational scenarios (e.g., streaming chatbots), the primary bottleneck is often the **playback or reading speed** of the user, rather than the raw generation speed. Given the flexibility GOOD offers for black-box alignment, we believe the method remains highly valuable in these speed-insensitive scenarios where the slight overhead of guidance transmission does not negatively impact the user experience.
>
> ## To Q3: Compute Accounting and Net Speedups
>
> The "Net Speedup" is achieved through the mechanism of **Speculative Decoding**.
>
> - **Cost:** We do run three models. However, Models A and $A_{it}$ are typically distinctively smaller than Model B (e.g., 2B vs. 70B). The computational cost of running the small models is a fraction of running the large model.
> - **Gain:** In standard decoding, Model B runs 1 forward pass per token. In GOOD (speculative), Model B runs 1 forward pass to verify a *batch* of $K$ tokens. If the acceptance rate is high, we generate $K$ tokens for the cost of $\approx 1$ pass of B.
> - **Scaling:** Alignment interventions usually occur at specific stylistic points. For the majority of the generation where A and $A_{it}$ agree (no alignment needed), GOOD functions effectively as a 2-layer speculative decoding system, enjoying the associated speedups.

---

> ### Author Response · Authors · 2025-12-02
>
> We hope these responses clarify the mechanics and practicality of GOOD. We are committed to incorporating these clarifications into the final manuscript.

---

### Author Response · Authors · 2025-12-02
**Summary of Rebuttal and Response to Reviewer Concerns**

**Dear Area Chairs,**

We sincerely thank the ACs and reviewers for their dedicated time and effort in evaluating our work, especially given the technical challenges faced by the conference platform this year. We value the constructive feedback and have carefully addressed all concerns in our individual responses.

**1. Consensus on Strengths**

We are encouraged that the core contributions of GOOD were recognized by multiple reviewers:

- **Practicality:** Reviewers **xzpM**, **FQxu**, and **uhDe** validated our method’s capability to work in practical black-box settings. Reviewer **xzpM** specifically noted the API design is "useful for practitioners," and **uhDe** praised the method as "originality … is very high" and "highly creative."
- **Effectiveness:** **FQxu** and **uhDe** acknowledged that our experiments on alignment performance are extensive and convincing. The other two reviewers also raise no doubts about this.
- **Efficiency Mechanism:** Both **xzpM** and **uhDe** commended our strategy of folding alignment guidance into a **speculate-and-verify loop**, deeming it a smart and "essential" integration for making online intervention practical.

**2. Resolution of Key Concerns**

The primary reasons for the borderline scores (4) stem largely from verifiable misunderstandings or concerns that have been addressed by our clarifications and are actually recognized as strengths by other reviewers:

- Clarification on "Black-Box" Access (Addressing xzpM & uhDe):

    The main reservation from reviewers xzpM and uhDe was the misconception that GOOD requires access to the target model's logits or vocabulary. As correctly identified by mpLJ and FQxu, our method operates strictly at the string level for the target model. We have clarified this mechanism in our responses and will revise the method description in the final version to eliminate this ambiguity. With this misunderstanding resolved, the premise for their concern is removed.

- Justification of Motivation and Cost (Addressing mpLJ):

    Reviewer **mpLJ** was concerned that introducing extra guiding models incurs unnecessary cost. We clarified that using smaller models for **Speculative Decoding** is already a standard acceleration practice. GOOD does not waste resources; rather, it upgrades this necessary "drafting" step to perform alignment simultaneously. This innovative combination of alignment and acceleration was explicitly recognized as a core strength by **xzpM** and **uhDe**.

- Specific Inquiries (Addressing FQxu):

    We have provided detailed responses to FQxu's questions regarding method naming ("Optimal" as an objective), hardware consistency in testing, and baseline selections.


**3. Future Work and Final Polish**

We agree that the reviewers raised insightful questions for future exploration, such as **uhDe**'s suggestion for a dedicated deep-reasoning benchmark. We are committed to incorporating these into our future work. Regarding the additional efficiency queries (e.g., speedup scaling with output length, intervention frequency thresholds), we commit to including these analyses in the appendix of the final version. We believe these details further characterize the method but do not alter the main conclusions and contributions.

**Conclusion**

While the uniform score of 4 indicates a borderline status, we respectfully submit that the primary drivers for these scores were either specific misunderstandings (now clarified) or concerns regarding motivation that contradict the positive consensus of other reviewers.

GOOD represents a timely solution for aligning closed-source models without parameter access, balancing performance with efficiency. We earnestly hope the Area Chairs will consider the clarifications provided during the rebuttal and the potential of this work to advance practical LLM alignment.

**Sincerely,**

**The Authors**

---

### Meta-Review · Area_Chair_rsAf · 2026-01-04

**Summary:**

All the original reviews are not positive and they raised many issues related to significance, novelty and evaluation. The author response (which is quite detailed) addressed some of concerns. But overall, the quality of the paper seems slightly below the bar of ICLR.

**Reviewer Concerns:**

The authors have gave detailed response to each question. Some concerns are not fully addressed such as uhDe's suggestion for a dedicated deep-reasoning benchmark.

**Reviewer Scores:**

Most reviewers will slightly increase their scores.

---

### Decision · Program_Chairs · 2026-01-26

Reject